# OpenReview forum: "Nearly Optimal Algorithms for Contextual Dueling Bandits from Adversarial Feedback"
_ICML.cc/2025/Conference — ICML 2025 poster_

### Official Review · Reviewer_X3W6 · 2025-03-08

**Overall Recommendation:** 3

**Summary:**

This paper addresses the problem of adversarial attack in contextual dueling bandits. The authors propose a new algorithm (RCDB) that integrates uncertainty-weighted maximum likelihood estimation to mitigate the impact of adversarial feedback. They obtain a near-optimal regret bound that is robust to adversarial feedback. Moreover, the authors develop an enhanced variant (RCDB-S) that eliminates exponential dependencies. Empirical evaluations confirm the superiority of the proposed methods over existing algorithms under various adversarial conditions.

**Claims And Evidence:**

The overall claims in the paper are clear and well-supported by both theoretical analysis and empirical results.

**Essential References Not Discussed:**

I think the paper covers the related work quite thoroughly.

**Experimental Designs Or Analyses:**

The empirical results for the second algorithm (RCDB-S) are missing. It would be helpful if the authors included experiments to clearly demonstrate the practical advantages and performance improvements of RCDB-S compared to RCDB and other existing methods.

**Methods And Evaluation Criteria:**

Overall, the proposed methods are well-suited to the addressed problem setting.
However, there are practical concerns since the algorithm requires knowledge of both parameters $C$ and $\kappa$, which are typically hard to determine in practice. The authors do discuss scenarios with an unknown number of adversarial attacks; however, if the true $C$ exceeds the adversarial tolerance threshold, performance can degrade significantly. Moreover, the first algorithm (RCDB) is  highly rely on the method proposed in He et al. 2022.

On a positive note, the second proposed method (RCDB-S) is particularly interesting, as it is the first to eliminate the dependency on $\kappa$ in the regret bound for contextual dueling bandits.

**Other Comments Or Suggestions:**

No other comments.

**Other Strengths And Weaknesses:**

No other comments.

**Questions For Authors:**

1. Can the authors provide empirical results for RCDB-S? How does its practical performance compare with RCDB and other baseline methods?

2. Are there practical methods or heuristics to estimate or approximate the adversarial corruption level $C$, instead of assuming it's known in advance?

3. If the regret definition is changed to "weak regret," is it still possible to eliminate the $\kappa$-dependence from the regret bound?

**Relation To Broader Scientific Literature:**

I find the $\kappa$-independent regret bound particularly intriguing, as it represents a significant theoretical improvement and clearly differentiates this work within the broader literature on contextual dueling bandits.

**Theoretical Claims:**

The theoretical claims are well-reasoned, and the proofs appear to be correct

---

> ### Author Rebuttal · Authors · 2025-04-01
>
> Thank you for your positive feedback! We will address your concerns.
>
> **Q1**: Empirical results for RCDB-S
>
> **A1**: We will add the empirical results for RCDB-S in our revision. As an example, the performance under the greedy attack setting (as described in Section E.1) is summarized in the table below. Compared to RCDB, RCDB-S performs similarly or slightly worse in the early rounds (e.g., at t=500,1000). This is consistent with our theoretical analysis, where RCDB-S incurs a larger corruption term of  $\tilde O(dBC/\kappa)$ (along with some other lower order terms), compared with $\tilde O(dC/\kappa) $ of RCDB. As $t$ increases, RCDB-S gradually outperforms RCDB due to its improved dependence on the dominant term in $T$ of $\tilde O(d\sqrt{T})$.
> Theoretically, as discussed in Section 6.1, our algorithm relies on the local derivative, which starts close to the lower bound $\kappa$ and gradually increases toward a constant during the learning process. This creates a growing gap between the current local derivative and the initial lower bound $\kappa$. As the number of rounds $T$ grows, this gap widens, and the difference between the two methods becomes more significant. As a result, the regret growth of RCDB-S slows down significantly over time—for instance, between $t=1500$ and $t=2000$, its regret increases by only 11.1, compared to 64.9 for RCDB—demonstrating its increasingly improved performance in later stages.
> This further explains the improved performance as $t$ increases. Therefore, our experimental results are well aligned with the theoretical findings.
>
> | |  t = 500 | t = 1000 | t = 1500 | t = 2000|
> | ----------|------------|------------|------------|------------|
> RCDB |  547.0 | 642.2 | 713.8 | 778.7|
> RCDB-S|  556.4 | 642.7| 680.1| 691.2|
>
> ---
>
> **Q2**: Heuristics to approximate the adversarial corruption level
>
> **A2**: The adversarial corruption level inherently depends on the environment. In extreme cases, it can become arbitrarily large, inevitably causing any algorithm to fail. Therefore, attempting to approximate the corruption level $C$ precisely is generally not meaningful. A practical and theoretically justified heuristic, as discussed in Section 5.2, is to set $C = O(\sqrt{T})$. Adopting this heuristic choice ensures theoretical optimality, and our simulation experiments demonstrate its good empirical performance.
>
> ---
>
> **Q3**: Eliminate the $\kappa$-dependence for weak regret
>
> **A3**: We are not entirely clear on the definition of weak regret, as mentioned in your question. Could you please clarify its meaning? We will address your questions in the discussion that follows.

---

> > ### Comment · Reviewer_X3W6 · 2025-04-02
> >
> > Thank you for the detailed response and for including the additional experiment.
> >
> > Regarding the regret,  I referred to "weak regret" using the term used by the authors in Line 220: *"weak regret defined in Bengs et al. (2022), which only considers the reward gap of the better action."* Since the authors used this term, I think they are more familiar with its precise meaning. Could you discuss the possibility of achieving $\kappa$-free regret under the "weak regret"?
> >
> > Additionally, another issue came to mind upon revisiting the paper. The proposed lower bound appears to be in tension with the result of Theorem 6.1 ($\kappa$-free upper bound). In particular, since $1/\kappa$ can be exponentially large, the proposed lower bound may actually exceed the regret upper bound in Theorem 6.1. Could you clarify this contradiction? If I’m missing something, I’d appreciate your clarification.

---

> > > ### Author Response · Authors · 2025-04-03
> > >
> > > Thanks for your reply. We will address your questions!
> > >
> > > In Bengs et al. (2022), the weak regret is defined as $R_w(T) = \sum_{t} r^*( x_t, a_t^*) - \max_{ a \in \lbrace a_t, b_t\rbrace} r^*(x_t, a)$. In comparison, our regret is defined by $R(T) = \sum_{i} 2r^*( x_t, a_t^*) -  r^*(x_t, a_t) - r^*(x_t, b_t)$. It immediately follows that $R_w(T) \le R(T)$. Therefore, removing the $\kappa$ dependence from our regret bound under the sigmoid link function directly yields the same improvement for the weak regret. Moreover, under the more general Assumption 3.2 (see discussion below), our argument in Theorem 5.4 regarding $\kappa$ dependence extends to the weak regret as well, suggesting that $\kappa$ cannot be eliminated in that case either.
> > >
> > > Regarding the second question, we emphasize that our lower bound applies to a broad class of link functions, subject only to **Assumption 3.2**. Theorem 3.2 demonstrates that, for a general link function, if an algorithm only has access to the lower bound of the gradient \(\kappa\), it is impossible to achieve a regret lower than that of our **RCDB method**. However, for specific choices—such as the **sigmoid link function**—more refined algorithms that exploit the dynamics of the link function can lead to improved performance, as discussed at the beginning of **Section 6**.

---

### Official Review · Reviewer_Jfpo · 2025-03-11

**Overall Recommendation:** 4

**Summary:**

The paper considers the problem of average regret minimization in stochastic contextual dueling bandits under the linear transitivity model and *strong* corruption. The authors first propose RCDB, a robustified version of MaxInP (Saha, 2021) that computes a weighted MLE, and proves its regret bound of $\widetilde{O}(d(\sqrt{T} + C)/\kappa)$, where $\kappa$ is some curvature-dependent, problem difficulty quantity. This regret holds when $C$ is known or unknown; in the latter case, $C$ can be replaced with $\overline{C}$, an adversarial tolerance threshold. This is complemented with a lower bound that holds for a piecewise linear link function, for which the optimality of RCDB in $d, C, T, \kappa$ follows. When the link function is sigmoidal, the authors propose RCBD-S, an improved version of RCDB that more effectively utilizes the local derivative information of sigmoids. This leads to an improved guarantee of $\widetilde{O}(d \sqrt{T} + dC/\kappa + d^2 / \kappa^2)$, where the leading term is $\kappa$-free, the first result to show such improvement in contextual dueling bandits literature. The efficacy of the algorithms is shown numerically as well.

**Claims And Evidence:**

See my comments on Theoretical Results and Experiments

**Essential References Not Discussed:**

To my knowledge, no *essential* references were left out.

Although slightly different, please consider discussing Jun et al. (2021), where a $\kappa$-free confidence bound for $|\langle x, \hat{\theta} - \theta_\star \rangle|$ was proved under a fixed design. This seems relevant to the $\Lambda_t$-based confidence sequence that the authors prove in Appendix C (event $\mathcal{E}_2$).


https://proceedings.mlr.press/v139/jun21a.html

**Experimental Designs Or Analyses:**

The experimental designs seem appropriate. Some minor comments:
1. Figure 2 is never referenced explicitly in the text.
2. Error bars missing from Figure 2.

**Methods And Evaluation Criteria:**

See Experimental Designs or Analyses

**Other Comments Or Suggestions:**

**Suggestions:**
1. I think it would be better to move footnote 1 to the main text.
2. Although all relevant citations are included, I still believe that they should be appropriately cited when needed. For instance,
   - At the end of Section 4, the authors should mention that the precise form of estimated reward + exploration bonus for dueling bandits is basically MaxInP of Saha (2021)
   - The Taylor expansion argument combined with self-concordance (e.g., Section 6.1), as well as the self-bounding equation (e.g., last part of the proof of Theorem 6.1), were pioneered by Abeille et al. (2021).
   - It seems that the overall proof flow of Theorem 5.4 resembles the lower bound proof of Li et al. (2024)
3. There is no mention of experiments nor its location in the main text. The authors should mention somewhere that they have provided extensive numerical experiments in Appendix E.
4. Theorem 5.5 (and Sec 5.2 in general) seems redundant with Section 5.1.
5. It would be helpful to put in the citation/exact reference for the averaging hammer in Line 792, e.g., Section 24.1 of Lattimore & Szepesvari (2020).
6. Another interesting future work that I ask the authors to consider putting in would be making the algorithm more efficient in the sense that MLEs do not get computed at every iteration [1,2,3] or use hashing [4].
7. For the proof of the lower bound, the authors define a stopping time $\tau_i$ and some function $U_{\theta,i}(x)$ similar to Li et al. (2024). Some intuition would be nice on what these two quantities mean. I understood $\tau_i$ as the first time $\tau$ in which the amount of information gathered for coordinate $i$ til time $\tau$ (quantified as the sum of squares of the $i$-th coordinates of the chosen arms) exceeds the average? amount of information expected, which is $2T/d$ ($T/d$ per coordinate times 2). I understood $U_{\theta,i}(x)$ as the lower bounding term that pops up when lower bounding the average dueling regret. Are my intuitions correct?


**Typos:**
1. Line 231 (right column): What are the "first" and "last" inequalities?
2. Line 233 (right column): $\geq$ => $\succeq$
3. Line 965: equalties => equalities


[1] https://openreview.net/forum?id=FTPDBQuT4G

[2] https://openreview.net/forum?id=ofa1U5BJVJ

[3] https://proceedings.mlr.press/v151/faury22a.html

[4] https://papers.nips.cc/paper_files/paper/2017/hash/28dd2c7955ce926456240b2ff0100bde-Abstract.html

**Other Strengths And Weaknesses:**

**Strengths:**
1. Clearly and well written
2. The dependencies on all the problem-dependent quantities are well-tracked. This includes $d, T, \kappa, B$.
3. Nontrivial combination of ideas and techniques from logistic bandit (Abeille et al., 2021), adversarial bandits (He et al., 2022), and dueling bandits (Saha, 2021), especially RCDB-S where several (new) properties of sigmoid were used in conjunction with the mentioned techniques

**Weaknesses:**
1. Strength #3 is also partially a weakness, but not a huge one.

**Questions For Authors:**

1. In logistic and GLM bandit literature, the "correct" kappa is $\kappa_\star = \frac{1}{\dot{\sigma}(\langle x_\star, \theta_\star \rangle)}$. Indeed, to me, both the upper and lower bounds for the considered dueling bandits setting should depend on something like $\kappa_\star = \max_{x \in \mathcal{X}} \max_{a, b \in \mathcal{A}} \frac{1}{\dot{\sigma}\left( r^*(x, a) - r^*(x, b) \right)}$. Given that the proofs utilize Taylor expansion of $\sigma(\phi^\top \theta)$ about $\theta = \theta_\star$ as in Abeille et al. (2021), I strongly believe that the current analysis can be (somewhat easily) improved. If I am missing something, please let me know!
2. At the end of Section 4, what is the authors' intention for referring the reader to Appendix A of Di et al. (2023) in the current context? Appendix A of Di et al. (2023) seems to be regarding the layered version of MaxInP for variance-aware regret guarantees. Is it to allude that a similar variance-aware guarantee can also be obtained in the adversarial dueling setting? If this is the intention, it should be made more explicit.
3. I am curious about whether one can use similar techniques from Abeille et al. (2021) to show a *local* minimax lower bound, i.e., given some instance $\theta_\star$, there exists another $\theta_\star'$ in its neighborhood such that .... Similarly, I wonder whether one can show a local minimax lower bound of $\Omega(d \sqrt{T} + ...)$ for sigmoidal $\sigma$. These two points would have made the paper much stronger in my opinion.
4. The proof of RCDB-S seems to use properties highly specific to sigmoid (e.g., Line 1165-1172, Line 1210-1220). In general, can the similar algorithm principle and analysis be extended to $\sigma : \mathbb{R} \rightarrow [0, 1]$ is a link function that is 1. monotone increasing, 2. $\sigma(z) + \sigma(-z) = 1$, and 3. self-concordant (i.e., $|\ddot{\sigma}| \leq R_s \dot{\sigma}$)?
5. For constructing the weighted confidence sequence, could one obtain further improvements in factors of $B$ via the likelihood-based confidence sequences of [1,2,3]? If yes, I would suggest including this as a potential future direction as well.


[1] https://openreview.net/forum?id=4anryczeED

[2] https://proceedings.mlr.press/v238/lee24d.html

[3] https://openreview.net/forum?id=MDdOQayWTA

**Relation To Broader Scientific Literature:**

- To the best of my knowledge, it tackles a new problem setting of contextual dueling bandits with adversarial corruptions.
- Integrates three lines of works (logistic bandit, dueling bandit, bandits with adversarial corruption)

**Theoretical Claims:**

I checked all proofs and concur that they are generally correct. There are very minor issues that I list below, but they do not harm the theoretical claims in any way:
1. Line 666: $\leq$ => $=$
2. Line 687: $0$ if $x < -1/2$
3. Line 735: missing $1/2$ in the $\sqrt{}$ from Pinsker's inequality

Also, the lower bound (Theorem 5.4) and the following paragraph should be written more clearly to avoid any misunderstanding. The lower bound is only for the specific form of piecewise-linear linear function, and thus, RCDB is (minimax) optimal *only* for that link function.

---

> ### Author Rebuttal · Authors · 2025-04-01
>
> Thank you for your positive feedback! We will address your questions one by one.
>
> **Q1**:  Typos and suggestions
>
> **A1**: Thank you for pointing these out. We will address them in our next revision.
>
> ---
>
> **Q2**: The writing of lower bound
>
> **A2**: There seems to be a misunderstanding regarding our claim about the lower bound. Our established lower bound works for the general class of link functions constrained only by Assumption 3.2. Within this general class, we construct a piecewise-linear example to establish the lower bound of $\Omega((d\sqrt{T} + dC)/\kappa)$. This bound matches the upper bound achieved by our RCDB algorithm, which operates under the same general Assumption 3.2. Consequently, our algorithm is minimax optimal in this general scenario. As explicitly noted at the beginning of Section 6, when we move beyond the general assumption, focusing on specific link functions (e.g., sigmoid function), we show that it is possible to achieve improved dependency on $\kappa$.
>
> ---
>
> **Q3**: Relation with Jun et al. (2021)
>
> **A3**:  Jun et al.(2021) considered the logistic bandit problem with pure exploration and proposed an algorithm with a sample complexity guarantee. In contrast, our work focuses on the regret of dueling bandits, with the approach involving reward estimation complemented by a bonus term.
>
> Furthermore, Jun et al. (2021) derived a concentration inequality under a fixed design assumption, given by $|\langle x, \hat{\theta} - \theta^* \rangle|\leq\beta||x|| _ {H_t(\theta^*)^{-1}}$. However, this assumption is too restrictive for our setting with adaptive action selection. Therefore, we establish a concentration inequality applicable to adaptive designs that holds for any arm $x$ ($\mathcal{E} _ 2$ in Line 1054), which works for the adaptive arm-selection inherent to bandit algorithms.
> We will explicitly discuss this comparison with Jun et al.. (2021) in the revised version.
>
> ---
>
> **Q4**: The combination of ideas and techniques from logistic bandits, adversarial bandits, and dueling bandits is also a minor weakness
>
> **A4**: We'd like to mention that besides integrating concepts from logistic bandits, adversarial bandits, and dueling bandits, we propose a novel idea of a well-constructed weight $v_t$ (Line 15 of Algorithm 2), representing a significant contribution. For a detailed discussion, please refer to **A2** to Reviewer TxSJ due to space constraints of the rebuttal.
>
> ---
>
> We will next address your ''questions for authors'' part.
>
> **Q5**: Correct $\kappa$
>
> **A5**: The "correct" $\kappa$ for logistic bandits is exactly given by $\kappa^*=1/\dot \sigma(x _ * ^\top\theta ^*)$, where $x_ * =\text{argmax }x^\top\theta^*$. As the analogy, the corresponding $\kappa$ in the dueling bandit setting should be $\kappa^* = 1/\dot \sigma(x_* ^\top \theta^*-x _ * ^\top\theta^*) $, with $x_* = \text{argmax } x^\top \theta^*$. It is a constant and it's exactly what we use. This difference stems from the nature of dueling bandits, which require **both** arms to approach optimality, unlike standard logistic bandits that involve only one arm. Indeed, if we define $\kappa^*$ explicitly as $\max_x \max_{a,b}1/\dot \sigma((r^*(x,a) - r^*(x,b))$, this corresponds precisely to the inverse of the $\kappa$ presented in Assumption 3.2, leading to a worse regret bound similar to that of Theorem 5.3.
>
> ---
>
> **Q6**: Referring the reader to Appendix A of Di et al. (2023)
>
> **A6**: We apologize for the earlier typos. We intended to refer to Appendix C of Di et al. (2023), which discussed different arm-selection rules involving the selection of two arms. As this aspect has already been studied and is not the central contribution of our algorithm, we have included a reference for readers seeking a detailed discussion.
>
> ---
>
> **Q7**: Local minimax lower bound
>
> **A7**: It is possible to establish a local minimax lower bound. In our current proof, we consider the parameter set $\Theta = \lbrace -\Delta, \Delta \rbrace^{d}$ (Line 693). If we instead focus on a local parameter class such as $\Theta = \theta^* + \lbrace -\Delta, \Delta \rbrace^{d}$, similar to the approach in Abeille et al. (2021), we believe that a local minimax lower bound can be derived.
>
> ---
>
> **Q8**: Extend the analysis to self-concordant link function
>
> **A8**: We believe our analysis can be extended to the self-concordant setting, although this extension requires some additional nontrivial effort. Specifically, regarding the part mentioned by the reviewer where we apply properties of the logistic function, we could instead leverage a Taylor expansion to achieve similar results. For example, in lines 1210–1220, we can derive $\frac{1}{\dot \sigma(\hat \Delta_t)} \le \frac{1}{\dot \sigma(0)} + \int_0^{\hat \Delta_t} \frac{|\ddot \sigma(s)|}{|\dot \sigma^2(s)|} ds$. Then, utilizing the self-concordance property, we can bound the integral term by $\hat\Delta_t R_s/\kappa$. allowing the subsequent steps in our proof to proceed similarly.

---

> > ### Comment · Reviewer_Jfpo · 2025-04-07
> >
> > Thank you for the detailed responses, which answered most of my questions/concerns. Also, apologies for the late rebuttal comment from my side. I intend to keep my score, leaning towards acceptance. Although there isn't much time, a few more would give me further clarification and potentially a higher score.
> >
> > ----
> >
> > **Questions**
> >
> > 1. After some thoughts, I now fully understand the correct $\kappa$. But then, I feel that as the "correct" $\kappa$ is $1$ for any $\sigma$ with $\sigma(0) = 1$, the leading term should be free of $\kappa^{-1}$ and it should only be the transient ($\sqrt{T}$-free) term that depends on $\kappa^{-1}$. This is because after paying $\kappa^{-1}$ dependent cost in the beginning, as the algorithm should've found the $\theta_\star$ quite accurately, by linear approximation, the leading term should be free (or even benefit) from the nonlinearities. Or is my intuition wrong somewhere? This, again, stems from my understanding of logistic/GLM bandits, and so I may be missing something that is crucial in dueling bandits.
> >
> > 2. Continuing, in the weak regret as mentioned by reviewer X3W6, as the weak regret does not have a "dueling nature" explicit in its definition, is it possible that the regret can be $\kappa$-free? I know that the authors responded that the same intuition holds for the weak regret as well, but then, is there any chance that the inequality weak <= strong is loose?
> >
> > 3. Is my understanding of the lower bound proof as presented in Suggestion 7 correct? If so, it would be nice for the authors to include this intuition in the Appendix.
> >
> >
> > **Suggestions**
> >
> > 1. I do *not* expect this to happen by the end of the rebuttal phase, but it would be better if the authors could work on further improving the guarantees that are deemed possible, especially my Q7 and Q8.
> > 2. As mentioned in my Q3, it would be very cool (and add a lot to the technical novelty of the paper) if the authors could derive a $\kappa$-free regret lower bound as well for sigmoidal $\sigma$! I have a feeling that something similar to Abeille et al. (2021) may do the trick? Also, this I do *not* expect to happen by the end of the rebuttal phase.
> >
> >
> >
> > -----
> > -----
> > **After authors' second rebuttal**
> >
> > I sincerely thank the authors for the enlightening discussions and for providing satisfactory answers. As all of my concerns and questions have been addressed (despite the lack of time), I am raising my score. I would like the authors to include all the relevant discussions (at least in the Appendix if the space doesn't allow), as all of these would be of great interest to the bandits community (especially for those working on logistic/GLM bandits like me).

---

> > > ### Author Response · Authors · 2025-04-08
> > >
> > > Thank you for your questions. We'd be glad to discuss these questions.
> > >
> > > **Q1**: More discussion on the correct $\kappa$
> > >
> > > **A1**: Your intuition from the logistic/GLM literature is definitely correct. There is a small typo: the "correct" $\kappa$ depends on $\dot \sigma(0)$, which is always 0.25 for the sigmoid function. What you said is exactly what we want to present in Theorem 6.1, a $\kappa$ independent leading term $O(dB^{1.5} T)$, plus some $\kappa$ dependent transient terms. Note that this improvement can only be made for the logistic function (or more generally, self-concordant functions). Therefore, we cannot achieve a similar result in Theorem 5.3 when we consider a general link function.
> > >
> > > ---
> > >
> > > **Q2**: Discussion about the local lower bound in Abeille et al. (2021)
> > >
> > > **A2**: Using the first part of the proof of Theorem 5.4 (or by referring to Li et al., 2024 as we mentioned), we can directly get a $\kappa$-free lower bound $\Omega(d\sqrt{T})$ for the sigmoid function. In fact, when $\kappa < 1$, our $\Omega(d\sqrt{T}/\kappa)$ in Theorem 5.4 is strictly tighter than the $\kappa$-free lower bound.
> > >
> > > We understand that your concern stems from Abeille et al. (2021), where a $\kappa$-dependent lower bound is tighter than $\kappa$-free ones, which seems in contrast to our result. The key difference lies in the reward structure: in their setting, the reward function is **nonlinear**, leading to a regret expression of $\text{Regret}(T) = \sum \mu(x^ {* \top} \theta^*) - \mu(x_ t ^\top \theta^*) = \dot \mu(x^ {* \top} \theta^*)  ((x ^*-x_ t)^\top \theta^*) + \ldots$. In contrast, our setting assumes a **linear** reward, resulting in a regret of $\text{Regret}(T) = ((x^*-x_ t)^\top \theta^*)  + ((x^*-y_t)^\top \theta^*)$. The existence of the additional $\dot{\mu}$ term allows them to get a regret bound $O(d\sqrt{\kappa ^* T})$, which benefits from the curvature of the nonlinear function. In our setting, on the contrary, without the additional $\dot \mu$, we can intuitively expect the correct regret bound to be $O(d\sqrt{T / \kappa ^* })$. As we have discussed, the correct $\kappa$ in our setting is a constant. Therefore, it will only incur an additional constant factor in our setting
> > >
> > > While we find the local minimax lower bound in Abeille et al. (2021) to be an interesting setting, we do not think it leads to improved rates in our setting. Specifically, it would only introduce a constant factor related to $\dot \sigma(0) = 0.25$, as opposed to the instance-dependent $\dot \mu(x^ {*\top} \theta^ *)$ in their setting. A similar situation arises in the upper bound. As we discussed in the last paragraph, the correct $\kappa$ is a constant, which is also omitted in our $O(d\sqrt{T})$ upper bound. For this reason, we did not include this discussion in our paper.
> > >
> > > ---
> > >
> > > **Q3**: The weak regret
> > >
> > > **A3**: Our previous argument shows that for weak regret, a matching $O(1/\kappa)$ dependence arises in both the upper and lower bounds under a general assumption on the link function. The lower bound comes from a similar argument as Theorem 5.4. And the upper bound comes from the inequality "weak <= strong".
> > >
> > > "Is there any chance that the inequality weak <= strong is loose?"
> > >
> > > We understand that this regards the special case of the sigmoid link function. As you pointed out, weak regret does not have a dueling nature, so it is possible that it could depend on a different $\kappa$ than $\dot\sigma(0)$. However, as discussed in **A2**, we cannot benefit from nonlinearity in our setting. Thus, the best choice of $\kappa$ is still $\dot\sigma(0)$, which is the maximum of $\dot\sigma$. In conclusion, even for weak regret, the correct dependence on $\kappa$ in the regret bound should be $\kappa$-free—just as in the case of strong regret.
> > >
> > > ---
> > >
> > > **Q4**: Lower bound proof
> > >
> > > **A4**: Yes, your understanding is correct. We will include this intuition in our revision.
> > >
> > > ---
> > >
> > > **Q5**: Self-concordance property.
> > >
> > > **A5**:
> > > Thank you for your suggestions. During the rebuttal process, we’ve been actively thinking about this interesting direction. In doing so, we’ve identified another potential issue that could significantly impact our algorithm design.
> > > In our current approach, we require the condition $\dot \sigma(\phi_i^\top \theta^*) \ge v_i (* )$  with high probability (Line 385), relying on the property $\ddot \sigma(x) \le 0$ when $x > 0$. For a general self-concordant link function, the property does not always hold. Thus, we may need to once again leverage Taylor's expansion and the self-concordance property to develop a new weight construction that still satisfies condition (*). While we are currently not sure what effect it will cause on the theoretical analysis, we will continue to explore this direction.
> > >
> > >
> > > ---
> > > We truly appreciate the insightful discussion with you and are grateful for your support in raising the score. We’ll ensure our discussion is included in the revised version.

---

### Official Review · Reviewer_t957 · 2025-03-14

**Overall Recommendation:** 3

**Summary:**

The paper considers the adversarial corruption setup in Dueling bandits and proposes an algorithm using the uncertainty-weighted maximum likelihood estimation and provides regret bounds and empirical evaluations.

**Claims And Evidence:**

Yes all theoretical claims have proofs and experimental results are provided.

**Essential References Not Discussed:**

All essential references have been discussed.

**Experimental Designs Or Analyses:**

The experimental designs are fair. However, they are on synthetic data and therefore are fairly limited and therefore the authors' description " We conduct extensive experiment" seems a bit exaggerated.

**Methods And Evaluation Criteria:**

Methods And Evaluation Criteria are fair.

**Other Comments Or Suggestions:**

-

**Other Strengths And Weaknesses:**

Weakness:

* The main contribution of the paper seems to be a direct extension of the algorithm developed in He et al 2022 for the linear bandit setup to the dueling bandit framework, uses the standard machinery from the dueling bandit literature, and as such lacks novelty.

**Questions For Authors:**

* Since this seems to be a direct extension of He et al 2022 using standard analysis techniques from the Duelling Bandits literature, could the authors describe novel aspects of their contribution in relation to existing literature.

* The contribution in Section 6 seems to be novel and provides tighter results for the Sigmoid Link function. Are there existing papers that analyze this special case for the non-corrupted case?

**Relation To Broader Scientific Literature:**

The contribution would be useful to the bandits/ sequential learning community.

**Theoretical Claims:**

None of the theoretical claims are proven in the main body of the paper.

---

> ### Author Rebuttal · Authors · 2025-04-01
>
> Thank you for your positive feedback! We will address your concerns.
>
> **Q1**: Direct extension of He et al. 2022 using standard analysis techniques from the Duelling Bandits
>
> **A1**: We believe that the reviewer overlooks several significant contributions in our work. First, we carefully analyze the dependency on $\kappa$ in Theorems 5.3 and 5.4. We introduce a new algorithm that achieves an $O((d\sqrt{T} + dC)/\kappa)$ regret bound. Furthermore, Theorem 5.4 establishes a matching lower bound, demonstrating that our result is optimal with respect to all involved parameters $d,T,C,\kappa$. Specifically, a $O(1/\kappa)$ dependency is both necessary and sufficient under the general assumption 3.2.
>
> Next, in Section 6, we illustrate how this dependency can be improved when considering specific link functions, such as the logistic function. To be more specific, through a more detailed analysis of the impact of $\kappa$ on the MLE, we identify the critical role of local derivatives and introduce a novel refined covariance matrix $\Lambda_t = \lambda I + \sum w_i v_i \phi_i \phi_i^\top$, where $v_i$ serves as an optimistic estimator of the local derivative (Lines 388-389). With our analysis, the "correct" weight ideally should be $\dot \sigma(\phi_i^\top \bar{\theta})$ (Line 348), where $\bar \theta$ is an intermediate value between $\theta^*$ and $\theta_t$. One might consider approximating this weight using either $\dot \sigma(\phi_i^\top \theta^*)$ or $\dot \sigma(\phi_i^\top \theta_t)$ directly. However, both direct approaches encounter critical issues: the first relies on the unknown parameter $\theta^*$, preventing us from applying the bonus term in Line 10 of Algorithm 2; the second fails due to the covariance matrix $\Lambda_t = \lambda I + \sum \dot\sigma(\phi_i^\top \theta_t) \phi_i\phi_i^\top$ depending on varying $\theta_t$, thus causing the matrix to not be monotonically increasing with $t$, posing significant analytical difficulties. Therefore, we propose constructing the weight $v_t$ as a carefully designed lower bound of the ideal weight. To the best of our knowledge, this specific technique has not appeared in prior logistic, adversarial, or dueling bandit literature.
>
> Finally, applying our new technique, we can remove the $\kappa$ dependency in the leading term of the regret upper bound, a result that has never been obtained in previous works for dueling bandits, even in the non-corrupted case. Thus, we firmly believe our contributions are substantial and that it is unfair to claim our work lacks novelty

---

### Official Review · Reviewer_TxSJ · 2025-03-14

**Overall Recommendation:** 3

**Summary:**

The paper studies the contextual dueling bandits with adversarial feedback, where a strong adversary may manipulate the preference label to mislead the agent, and the number of adversarial feedback is bound by $C$.
The authors propose an algorithm named RCDB to solve the problem. RCDB utilizes uncertainty-weighted maximum likelihood estimation (MLE) to reduce the effect of adversarial feedback.
The authors provide the regret upper bounds for both known and unknown $C$ and show the proposed regret bound is nearly minimax optimal with a lower bound.
The authors also present the RCDB-S algorithm for the sigmoid link function and provide an improved regret bound.

**Claims And Evidence:**

All claims are well-supported by clear and convincing evidence.

**Essential References Not Discussed:**

N/A

**Experimental Designs Or Analyses:**

The paper compares the proposed algorithm against several existing methods, such as MaxInP, CoLSTIM, and MaxPairUCB.
These comparisons are conducted using the cumulative regret metric, which is a standard metric in contextual bandits literature.
The relationship between cumulative regret and the number of adversarial feedback $C$ is evaluated.

**Methods And Evaluation Criteria:**

The proposed methods and evaluation criteria make sense for the problem.

**Other Comments Or Suggestions:**

N/A

**Other Strengths And Weaknesses:**

Strengths:

1. The problem of contextual dueling bandits with adversarial attacks is important.
2. Theoretical analysis shows that the regret is nearly minimax-optimal.
3. Empirical results show the performance of RCDB over existing dueling bandit algorithms in the presence of adversarial feedback.
4. The paper is well organized and all assumptions are clearly listed.

Weaknesses:

1. There is a mismatch between motivation and model. Although linear contextual dueling bandits are important and interesting, the authors motivate their setting by training LLM using RLHF. However, the reward function is assumed to be linear, which oversimplifies the complex, high-dimensional reward structures typically used in LLM training.
2. The paper builds heavily on the prior work of He et al. (2022), particularly in extending results to the **unknown** $C$, and using the argument that no algorithm can simultaneously achieve near-optimal regret when uncorrupted and maintain sublinear regret when  $C=\Omega(\sqrt{ T })$. This limits the contribution of this paper.
3. The weighted MLE to estimated $\theta$ has been studied by Di et al. (2023), where an auxilliary function are introduced.

**Questions For Authors:**

N/A

**Relation To Broader Scientific Literature:**

The paper is closely related to He et al. (2022) and Di et al. (2023). He et al. (2022)  study the problem of linear bandits with adversarial corruption, where weighted linear regression is used to deal with corruption, and Di et al (2023) study weighted maximum likelihood estimation.

**Theoretical Claims:**

I have not checked all the proofs in detail. I did not identify any obvious errors.

---

> ### Author Rebuttal · Authors · 2025-04-01
>
> Thank you for your feedback. We will address your concerns one by one.
>
> **Q1**: mismatch between motivation and model: the reward function is assumed to be linear, which oversimplifies the complex, high-dimensional reward structures typically used in LLM training
>
> **A1**:
> We believe that our motivation closely aligns with our contextual dueling bandit model, as the core challenge we address—effectively learning from preference feedback—is central to RLHF settings. Specifically, our model shares the same conceptual pipeline commonly used in RLHF: optimizing a reward function followed by applying a link function (such as the Bradley-Terry model) to derive preference scores. For clarity of representation, we focus on the linear reward function class, so as not to distract from our main contribution: addressing the challenges posed by adverarial preference feedback. This assumption is both standard and widely adopted in the literature [1–3].
> Moreover, it's straightforward to extend our techniques to more general nonlinear reward function classes [4][5]. We have discussed this in the future direction part.
>
> [1] Principled Reinforcement Learning with Human Feedback from Pairwise or K-wise Comparisons, Zhu et al. 2023, ICML2023
>
> [2] Iterative Preference Learning from Human Feedback: Bridging Theory and Practice for RLHF under KL-constraint Xiong et al. 2024 ICML2024
>
> [3]  Value-incentivized Preference Optimization: A Unified Approach to Online and Offline RLHF Cen et al.2025 ICLR2025
>
> [4] Corruption robust algorithms with uncertainty weighting for nonlinear contextual bandits and markov decision processes. Ye et al. 2023, ICML2023
>
> [5] Feel-good thompson sampling for contextual dueling bandits Li et al. 2024 ICML2024
>
> ---
>
> **Q2**: Relation with He et al. (2022) and Di et al. (2023)
>
> **A2**:
> First, Reviewer TxSJ claims that our work builds heavily on He et al. (2022), particularly in the extension to the unknown $C$ scenario, thus limiting our contribution. However, we clarify that this aspect represents only a minor section of our overall work. We incorporate this analysis, with explicit reference to He et al. (2022), primarily to demonstrate completeness to highlight that our proposed algorithm remains optimal even when $C$ is unknown, provided we select $\bar{C} = O(\sqrt{T})$. Including this discussion enhances, rather than diminishes, the significance and completeness of our work.
>
> Second, we acknowledge that weighted MLE has been previously explored by Di et al. (2023). However, we emphasize that both the selection of the weights and the intended purposes differ significantly from that work. Specifically, Di et al. (2023) select weights as $\alpha/||\phi_t|| _ {\Sigma_t^{-1}}^{2}$ to achieve a variance-aware regret bound. In contrast, our weight choice is $w_t = \min \lbrace 1, \alpha/||\phi_t|| _ { \Sigma_t^{-1}} \rbrace$ (equation 4.3, Line 235), strategically designed to cancel out the uncertainty. More precisely, our choice allows the equality $w_t ||\phi_t||_{\Sigma_t^{-1}} \alpha = 1$ when $w_t \le 1$ (Lines 663-672), effectively reducing the dependence on $T$ in the corruption term from $O(\sqrt{T})$ to $O(\log(T))$. This crucial difference highlights that our analytical goal and results are novel contributions distinct from those presented by Di et al. (2023).
>
> Last but not least, the reviewer overlooks several significant contributions of our work. To be more specific, through a more detailed analysis of the impact of $\kappa$ on the MLE, we identify the critical role of local derivatives and introduce a novel refined covariance matrix $\Lambda_t = \lambda I + \sum w_i v_i \phi_i \phi_i^\top$, where $v_i$ serves as an optimistic estimator of the local derivative (Lines 388-389). With our analysis, the "correct" weight ideally should be $\dot \sigma(\phi_i^\top \bar{\theta})$ (Line 348), where $\bar \theta$ is an intermediate value between $\theta^*$ and $\theta_t$. One might consider approximating this weight using either $\dot \sigma(\phi_i^\top \theta^*)$ or $\dot \sigma(\phi_i^\top \theta_t)$ directly. However, both direct approaches encounter critical issues: the first relies on the unknown parameter $\theta^*$, preventing us from applying the bonus term in Line 10 of Algorithm 2; the second fails due to the covariance matrix $\Lambda_t = \lambda I + \sum \dot\sigma(\phi_i^\top \theta_t) \phi_i\phi_i^\top$ depending on varying $\theta_t$, thus causing the matrix to not be monotonically increasing with $t$, posing significant analytical difficulties. Therefore, we propose constructing the weight $v_t$ as a carefully designed lower bound of the ideal weight. To the best of our knowledge, this specific technique has not appeared in prior logistic, adversarial, or dueling bandit literature.

---

> > ### Comment · Reviewer_TxSJ · 2025-04-04
> >
> > Thank you for the response.
> > 1. While linear models are standard in contextual bandits, my concern is about the appropriateness of using LLM training via RLHF as the main motivation. The issue is not mathematical tractability but a mismatch between the real complexity of RLHF and the simplified model in this paper. Citing a few theoretical works that use linear models does not resolve this disconnect. (There are more works that rely on complex, high-capacity models.)
> > Even though the authors mention that the method could be extended to nonlinear reward functions, it remains unclear how such an extension would be implemented both theoretically and practically
> > 1. I agree with the authors that the case when $C$ is unknown is a small part of the paper.
> > However, what the authors mention (such as (1) $w_t = \min \lbrace 1, \alpha/||\phi_t|| _ { \Sigma_t^{-1}} \rbrace$ to cancel out the uncertainty; (2) allows the equality $w_t ||\phi_t||_{\Sigma_t^{-1}} \alpha = 1$ when $w_t \le 1$) closely aligns with the previous work of He et al. (2022). Additionally, the challenges in the weighted maximum likelihood estimation (MLE) compared to the weighted LS (in He et al. (2022)) can be coped with techniques in Di et al. (2023).
> > The method heavily relies on He et al. (2022) and Di et al. (2023).
> > 1. I do find the techniques developed for handling the sigmoid link function to be novel and interesting. I suggest the authors emphasize the contribution of the algorithm for the sigmoid link function and corresponding technical novelties more. Because of this part, I will increase my score as I believe this component adds meaningful value despite the concerns raised above.

---

> > > ### Author Response · Authors · 2025-04-08
> > >
> > > Thank you for your reply. We will further elaborate on your concerns.
> > >
> > > 1. Extension to more general reward structures
> > >
> > >      At the core of our algorithm design is the careful design of the weight $w_t = \min \lbrace 1, \alpha/||\phi_t|| _ { \Sigma_t^{-1}} \rbrace$, and the construction of bonus term $||\phi_t|| _ { \Sigma_t^{-1}}$ (Line  6 in Algorithm 1) or $||\phi_t|| _ { \Lambda_t^{-1}}$ (Line  10 in Algorithm 2) in the dueling setting. Inspired by [1], we can generalize this bonus term using an uncertainty estimator of the form $\sup_{f_1,f_2} \frac{|f_1(\cdot)-f_2(\cdot)|}{\lambda + \sum_s |f_1(z_s)-f_2(z_s)|^2 * w_s }$. Since their formulation works for corruption-robust bandits with nonlinear function approximation, we believe our approach for dueling bandits can be similarly extended to the nonlinear setting.
> > >
> > > 2. We acknowledge that our proof of Theorem 5.3 builds upon techniques from He et al., and Di et al. This component is important to our presentation, as it introduces a state-of-the-art algorithm for dueling bandits with adversarial feedback that, to the best of our knowledge, has not previously appeared in the literature. Moreover, the analysis of the algorithm design motivates our discussion of the dependence on $\kappa$. Based on this, we have an improved algorithm design in the case of the logistic link function with an improved regret guarantee. Therefore, while the proof is a combination of the techniques in prior works, we believe their combination and application in our setting play a critical and original role in our overall presentation.
> > >
> > > 3. Thank you for recognizing our contributions related to the logistic link function and for increasing the score. We will make sure to highlight these aspects more clearly in our revised version.
> > >
> > > ---
> > >
> > > [1] Corruption robust algorithms with uncertainty weighting for nonlinear contextual bandits and markov decision processes. Ye et al. 2023, ICML2023

---

### Decision · Program_Chairs · 2025-05-01

**Decision:**

Accept (poster)

**Comment:**

The paper considers contextual dueling bandits with adversarial attacks. This is an important problem in the bandit literature. There were initially some concerns with parts of the paper but they were resolved during the author-reviewer rebuttal period and several reviewers raised their score accordingly, pointing to some novel techniques. Hence, I'm happy to recommend the paper to be accepted. Please take reviewers' comments into consideration when preparing the final version.